# Primary Feline Tauopathy: Clinical, Morphological, Immunohistochemical, and Genetic Studies

**DOI:** 10.3390/ani13182985

**Published:** 2023-09-21

**Authors:** Laura Vidal-Palencia, Cristina Font, Agustín Rebollada-Merino, Gabriel Santpere, Pol Andrés-Benito, Isidro Ferrer, Martí Pumarola

**Affiliations:** 1Hospital del Mar Research Institute, Parc de Recerca Biomèdica de Barcelona (PRBB), 08003 Barcelona, Spain; gsantpere@imim.es; 2Unitat de Patologia Murina i Comparada, Departament de Medicina i Cirurgia Animals, Facultat de Veterinària, Campus UAB, Travessera dels Turons s/n, 08193 Barcelona, Spain; marti.pumarola@uab.cat; 3Hospital Veterinari Canis, Can Pau Birol, 38, 17006 Girona, Spain; cfontn@hotmail.com; 4VISAVET Health Surveillance Centre, Complutense University of Madrid, 28040 Madrid, Spain; agusrebo@ucm.es; 5Department of Internal Medicine and Animal Surgery, Faculty of Veterinary Medicine, Complutense University of Madrid, 28040 Madrid, Spain; 6Department of Neuroscience, Yale University School of Medicine, New Haven, CT 06511, USA; 7Institut d’Investigació Biomèdica de Bellvitge (IDIBELL), 08908 Barcelona, Spain; pol.andres.benito@gmail.com (P.A.-B.); 8082ifa@gmail.com (I.F.); 8Department of Pathology and Experimental Therapeutics, University of Barcelona, 08007 Barcelona, Spain

**Keywords:** tauopathy, tau, cat, neurodegenerative disease, veterinary neuropathology

## Abstract

**Simple Summary:**

Tauopathies are a group of neurodegenerative diseases where a specific protein called tau accumulates and forms aggregates in neurons and glial cells. In humans, these diseases can be caused by only this protein (primary) or in combination with another one (secondary). Primary tauopathies are common in humans but rare in animals. We analyzed the development of tau pathology in 16 cats of different ages. A female cat showed progressive mental status and gait abnormalities over a six-year period. Brain imaging revealed a progressive shrinkage of the brain (atrophy). Due to a poor prognosis, the cat was euthanized at the age of ten years. Evaluation of the brain tissue showed significant loss of neurons in the parietal cortex and Purkinje cells in the cerebellum. Immunohistochemistry identified abnormal tau protein aggregates in neurons (referred to as pre-tangles) and oligodendrocytes (referred to as coiled bodies). Genetic testing did not reveal any known genetic alteration associated with this disease. None of the other 15 cats studied showed similar clinical signs or brain changes. This is the first reported case of primary tauopathy in an adult cat that presented the first neurological signs when she was four years old.

**Abstract:**

Tauopathies are a group of neurodegenerative diseases characterized by the pathological aggregation of hyperphosphorylated tau in neurons and glia. Primary tauopathies are not uncommon in humans but exceptional in other species. We evaluate the clinical, neuropathological, and genetic alterations related to tau pathology in 16 cats aged from 1 to 21 years with different clinical backgrounds. Interestingly, a 10-year-old female cat presented a six-year progressive history of mental status and gait abnormalities. The imaging study revealed generalized cortical atrophy. Due to the poor prognosis, the cat was euthanatized at the age of ten. Neuropathological lesions were characterized by massive neuronal loss with marked spongiosis and associated moderate reactive gliosis in the parietal cortex, being less severe in other areas of the cerebral cortex, and the loss of Purkinje cells of the cerebellum. Immunohistochemical methods revealed a 4R-tauopathy with granular pre-tangles in neurons and coiled bodies in oligodendrocytes. Deposits were recognized with several phospho-site antibodies (4Rtau, tau5, AT8, PFH, tau-P Thr181, tau-P-Ser 262, tau-P Ser 422) and associated with increased granular expression of active tau kinases (p38-P Thr180/Tyr182 and SAPK/JNK-P Thr138/Thr185). The genetic study revealed well-preserved coding regions of *MAPT.* No similar alterations related to tau pathology were found in the other 15 cats processed in parallel. To our knowledge, this is the first case reporting a primary 4R-tauopathy with severe cerebral and Purkinje cell degeneration in an adult cat with neurological signs starting at a young age.

## 1. Introduction

Tauopathies are clinical–pathological entities characterized by the disease-dependent intracellular deposition of abnormal tau protein in neurons and glial cells. Tau is a microtubule-associated protein (MAP) that maintains axonal transport and neuronal integrity, among other functions related to the cytoskeleton [1,2,3]. Tauopathies involve the modification of tau through post-translational processes, particularly hyper-phosphorylation. This leads to the formation of neurotoxic aggregates, characterized by a disease-dependent fibrillar structure and abnormal tri-dimensional conformation. These tau deposits are composed of tau species with three or four repeats (3Rtau, 4Rtau), resulting from exon 10 splicing of the gene encoding the microtubule-associated protein tau (*MAPT*) [2,4,5,6,7,8,9,10,11].

In humans, main tauopathies are categorized into six groups [12] (Figure 1) considering the anatomical distribution, the histological and cytological characterization, and the clinicopathological classification. The term primary tauopathies can also be used, in which tau inclusions are the predominant pathology, and secondary tauopathies that have tau inclusions, though they co-exist with another neuropathological hallmark [5,6,13]. However, in veterinary medicine, we are in the dark about the existence of such a diversity of tauopathies.

Intraneuronal accumulation of hyperphosphorylated tau has been reported in the brains of many mammalians, specifically dogs, cats, rabbits, sheep, goats, horses, donkeys, cheetahs, leopard cats, wolverines, brown bears, American bison, reindeer, degus, dolphins, and some primates [14,15,16,17,18]. In most cases, tau pathology is sporadic and usually restricted to clinically normal aged animals [15]. Interestingly, a rare example of a sporadic, naturally-occurring tauopathy in a non-primate species has been reported in aged cattle under the term idiopathic brainstem neuronal chromatolysis (IBNC). IBNC neuropathology does not correspond with any human tauopathy. Secondary accumulation of α-synuclein and ubiquitin was also present [19].

Aged cats may develop feline cognitive dysfunction (FCD; feline dementia) when older than 15 years and present β-amyloid deposition in the form of diffuse plaques rather than neuritic plaques. In a minority of cases, β-amyloid pathology is combined with tau pathology mimicking early-stage tangles (pre-tangles) rather than neurofibrillary tangles (NFTs) [20,21,22,23,24,25]. Tau deposits are variably distributed with contradictory results. Some studies have shown higher involvement of the hippocampus or the entorhinal cortex [20,22,23], rarely extending to the neocortex [23], and other studies have revealed the most significant number of pre-tangles in the cerebral cortex and fewer numbers in the entorhinal cortex and hippocampus [25]. These observations show that β-amyloid and tau pathology in aged cats is similar but not identical to that seen in human Alzheimer’s disease [23]. Moreover, a tauopathy without accompanying β-amyloid deposition was reported in two cats aged 19 years; in one of them, tau pathology prevailed in the hippocampus, temporal lobe, occipital cortex, and parietal cortex; and the other one was principally affected in the parietal cortex [24]. Regarding tau isoforms, 3R + 4Rtau has been reported in one study [22], whereas 4Rtau was the only isoform in the cases without concomitant β-amyloid deposition [24]. No tauopathies have been reported in cats with the clinical signs starting at a young age.

Here, we investigated the presence of tau pathology in the brains of 16 cats aged from 1 to 21 years, assessing the clinical, neuropathological, and genetic aspects. In one cat aged 18 years, we found a moderate tauopathy characterized by neurofibrillary pre-tangles in the hippocampus associated with β-amyloid diffuse plaques in the neocortex. Interestingly, another cat, 10 years old, presented a slowly progressive history (6 years) of loss of balance and difficulty walking that progressed to abnormal mental status and non-ambulatory tetraparesis. The post-mortem examination revealed a unique primary 4R-tauopathy accompanied by severe degeneration of both cerebral cortex and Purkinje cells in the cerebellum.

## 2. Materials and Methods

### 2.1. Animals and Control Samples

A total of 16 feline brains aged from 1 to 21 years and distinct neurological symptoms were retrospectively evaluated to study the development of tau pathology. Clinical data was acquired from the clinicians and veterinary facilities responsible for caring for the animals. Once they died or were euthanized, they were subsequently donated to the Unit of Murine and Comparative Pathology, Autonomous University of Barcelona, Spain. Owner agreements were obtained at the time of the animal’s donation. Post-mortem examination and the collection of the brains for histopathological study were performed 24 h after death. Formalin-fixed paraffin-embedded coronal sections were preserved until the time of the study. Representative sections of the brain were evaluated, including the frontal cortex/corpus striatum, parieto-temporal cortex/diencephalon, mesencephalon, cerebellum/pons, and medulla oblongata. All control samples were processed in parallel.

Positive control samples of tau pathology used to validate the experimental protocol were obtained from Alzheimer’s disease (AD) human patients in stages IV–VI (*n* = 3) and one canine brain. Brain human hippocampal complex samples were obtained from the Institute of Neuropathology brain bank, now incorporated into the HUB-ICO-IDIBELL Biobank. Ethical approval was obtained through the local ethics committee (CEIC) of the Bellvitge University Hospital, following the guidelines of Spanish legislation on this matter. The additional positive control brain sample from a 13-year-old dog was acquired from the Unit of Murine and Comparative Pathology, Autonomous University of Barcelona, Spain. Written consent for necropsy and histopathological analysis was obtained from the animal owner.

### 2.2. Magnetic Resonance Imaging (MRI)

Magnetic Resonance Imaging (MRI Esaote) was performed using a low-field 0.25-T (Vet-MR; Esaote) under general anesthesia. The study included T2-weighted and pre- and post-contrast (Gadoteridol: Dotarem at a dosage of 0.1 mMol/Kg), T1-Weighted images in dorsal, sagittal and transverse planes, and fluid-attenuated inversion recovery (FLAIR) images in transverse planes.

### 2.3. Histology and Immunohistochemistry (IHC)

The brains of all cats were removed from the skull and fixed in 10% buffered formalin. Serial coronal sections were embedded in paraffin. Sections 4 microns thick were obtained with a sliding microtome and mounted on glass slides. De-washed sections were stained with hematoxylin and eosin (HE), periodic acid Schiff (PAS), and Luxol fast Blue-Nissl (Klüver-Barrera staining, K-B) or processed for immunohistochemistry (IHC).

For IHC, endogenous peroxidases were blocked by incubation in 10% methanol-1% H_2_O_2_ solution (15 min) followed by 3% normal horse serum solution. Then, the sections were incubated at 4 °C overnight, or one hour at room temperature, with one of the primary antibodies (Table 1) with the antibody-dependent treatment of the sections. Following overnight incubation with the primary antibody, the sections were incubated with EnVision + system peroxidase (Dako, Agilent Technologies, Santa Clara, CA, USA) for 30 min at room temperature. The peroxidase reaction was visualized with 3,3′-diaminobenzidine (DAB) and H_2_O_2_. Sections were slightly counterstained with hematoxylin. Control of the immunostaining included omitting the primary antibody; no signal was obtained following incubation with only the secondary antibody.

We used antibodies against 3R tau, 4R tau, tau-P Thr181, tau AT8, and β-amyloid protein to analyze the tau pathology across all subjects. Following the initial suspicion of a primary tauopathy, we employed other antibodies to further investigate protein deposits in the brain sections of the suspected case (Table 1). An evaluation of the percentage of AT8-immunoreactive neurons for the total number of NeuN-immunoreactive cells was carried out in three parietal gyri: marginalis, ectomarginalis, and ectosylvius, and the temporal intersylvius gyrus. Counts were made separately in the upper layers (I–III) and inner layers (IV–V) in three same representative areas (0.140 mm^2^/1.49 mm) of consecutive sections stained with AT8 and NeuN antibodies.

### 2.4. Ultrastructural Study

Ultrastructural studies were performed on three cats, including the one with primary tauopathy (cat number 9) and the other two of a similar age (cat numbers 8 and 10). We studied the parietal cortex fixed in 10% buffered formalin, postfixed in 1% osmium tetraoxide, and embedded in EPON resin. Serial 1.5 mm semi-thin sections were stained with 1% toluidine blue and examined under a light microscope to study the overall organization of the cerebral cortex. Ultrathin (70 nm) sections were cut with a diamond knife, stained with lead citrate, and examined under a Hitachi H-7000 transmission electron microscope (EM) at 100 kV.

### 2.5. Sequencing, Read Mapping, and Variant Calling

Since only paraffin-embedded material was available for study, brain sections were dewaxed; DNA was extracted from these samples and processed for the genomic study. Pooled genomics capture libraries of the 16 cat brain samples were sequenced using MiSeq 300 cycles Micro v2 in two different runs. Reads were processed according to the recommendations of the xGen™ cfDNA & FFPE DNA Library Preparation Kit. We first trimmed all reads with trimmomatic-0.39 with the parameters SLIDINGWINDOW:4:25 and MINLEN:36. Fastq files were converted into BAM files using FastqToSam of PiccardTools 2.25. We then extracted UMIs from each read and incorporated them into the RX field in each BAM using ExtractUmisFromBam of the fgbio-2.1.0 package. After reconverting each BAM to fastq using SamToFastq from Piccard tools, we mapped reads to the felCat9 reference genome using BWA. We marked duplicates using UMIs with the Piccard tool MarkDuplicates and clipped overlapping reads using ClipBam from fgbio-2.1.0. BAMs derived from different runs on the same sample were merged using samtools. We calculate coverage statistics using the coverage from BEDTools/2.30.0.

We called variants in each sample using the GATK/4.0.1.2 HaplotypeCaller function and VarDictJava/1.8.3 around the targeted region of the *MAPT* gene, coordinates chrE1:46004718-46103101. We merged GATK individual GCF files using GenotypeGVCFs. Individual vcf files produced by VarDict were merged using the merge function of BCFtools/1.12. We annotated all variants in the VCF with Annovar. To do so, we created the required genome build using the Ensembl gene model of FelCat9, Felis_catus.Felis_catus_9.0.109.gff3. We compared our variants with those from the Ensembl variation database for FelCat9 release-109.

## 3. Results

### 3.1. Clinical Alterations Related to Tau Pathology

We conducted a clinical assessment on 16 cats aged 1 to 21 years with different clinical backgrounds. Most of the cats were affected by neurological disorders and either died or were euthanized following a veterinary evaluation that revealed further treatment would not improve their quality of life. None of the patients received a diagnosis of FCD, but three exhibited clinical signs consistent with a neurodegenerative disease (cats 9, 15, and 16; Table 2). Cat 9 was a female aged 10 years who showed a six-year history of progressively worsening mental status and gait abnormalities. Case 15, an 18-year-old Siamese cat, displayed behavioral alterations and acute neurological symptoms manifested as tonic and clonic seizures. Additionally, case 16 was a 21-year-old European cat that presented behavioral changes that could not be attributed to any other medical condition.

### 3.2. Tau Pathology in Cat Brains and Control Samples

Immunohistochemical labeling for tau protein revealed only 2 cats (numbers 9 and 15) of 16 showing positive immunoreactivity against phosphorylated tau (Table 2). Detailed post-mortem examination of cat 9’s brain tissue unveiled a primary 4R-tauopathy characterized by the presence of granular pre-tangles in neurons and coiled bodies in oligodendrocytes. Due to the unique and interesting case, we made a wide clinical and histopathological study detailed in the following subsection. On the other hand, case 15 presented a moderate secondary tauopathy characterized by neurofibrillary pre-tangles restricted to the hippocampus and, to a lesser extent, to the cerebral cortex and thalamus. Glial tau inclusions were absent. A few β-amyloid diffuse plaques were identified in the convexity of the cerebral hemispheres. β-amyloid angiopathy was absent.

Regarding the positive controls, large numbers of neurofibrillary tangles and senile plaques were observed in all cases with AD. The 13-year-old dog presented diffuse plaques of β-amyloid and pre-tangles in the neocortex.

### 3.3. Primary Feline Tauopathy

#### 3.3.1. Clinical History

A 7-year-old neutered female domestic European shorthair cat presented at the Neurology Service of the Canis Veterinary Hospital (Girona, Spain) due to a 3-year progressive history of loss of balance and difficulty when walking. The neurological examination revealed hypermetric ataxia of all four limbs, with bilateral episodes of loss of balance and intention tremor, consistent with a cerebellar neurolocalization.

Complete blood tests (including blood cell count, biochemistry, and FIV-FELV test) and chest and abdominal X-rays revealed no pathological changes. The cerebrospinal fluid (CSF) analysis was normal. The MRI study revealed mild brain atrophy with enlargement of the sulci and enlargement of the lateral ventricles. The cerebellum showed mild atrophy, and the fourth ventricle was enlarged (Figure 2A,B). No structural lesions or abnormal contrast uptake were observed.

The clinical signs slowly progressed. Three years later (6 years after the beginning of the clinical signs), the cat showed a severe disoriented mental status, non-ambulatory tetraparesis, hypertonicity of all four limbs, lack of menace response in both eyes, and episodes consistent with epileptic seizures. A new neuroimaging study was obtained in which we observed an asymmetrical and severe cerebral atrophy, mainly involving the parietal lobes and, to a lesser extent, the temporal lobes and hippocampus. The lateral ventricles were markedly enlarged, and the corpus callosum was abnormally thin. The fourth ventricle was also enlarged, and the cerebellum was severely atrophic (Figure 2C,D).

Due to the progression of the clinical signs and poor prognosis, the cat was euthanatized at the age of ten years.

#### 3.3.2. Macroscopic, Microscopic, and Immunohistochemical Examination

Macroscopically, the brain showed marked atrophy of the cerebral cortex and cerebellum and enlargement of all ventricles.

The microscopical examination of the cerebrum showed a bilateral and symmetric marked neuron loss and spongiosis in layers II, III, IV, and V, mainly affecting the paecruciatus gyri (frontal cortex) and the marginalis, ectomarginalis, and ectosylvius gyri (parietal cortex), with better preservation of the sulci [26] (Figure 3A,B and Appendix A). The remaining neurons presented small intracytoplasmic PAS-positive granules (Figure 3C), also stained with Klüver-Barrera (Figure 3D). These PAS-positive deposits also occurred in neurons of the rest of the neocortex and hippocampus and were interpreted as lipofuscin. Mild neuron loss was also observed in the cingulate cortex, temporal cortex, and hippocampal complex. The cerebellum showed a massive loss of Purkinje cells and a moderate loss of granule cells in the hemispheres and vermis (Figure 3E). Sections stained with calbindin D28K revealed the almost disappearance of Purkinje cells and the scarcity of dendritic arbors in the remaining cells (Figure 3F). In contrast, parallel sections stained with neurofilament antibodies decorated large numbers of empty baskets in the Purkinje cell layer (Figure 3G). All remaining structures of the telencephalon, including the entorhinal cortex, hippocampus, and brainstem, were well preserved on hematoxylin and eosin-stained sections.

The subcortical white matter of the parietal cortical area showed loss of myelin and an abundance of axonal swellings (Figure 3H and Appendix A), together with reduced numbers of olig2-immunoreactive cells (not shown). The corpus callosum was also atrophic, with reduced myelin and axons. Astrocytic gliosis, as revealed with anti-GFAP antibodies, was observed in all layers of the parietal cortex (Figure 3I) but was more marked in the temporal cortical area and hippocampus. Reactive astrocytes were also present in the subcortical white matter and corpus callosum. Reactive microgliosis was generalized but most abundant in the regions severely affected by neurodegeneration (Figure 3J). In the cerebellum, Bergmann glia increased in number, together with diffuse and mild microgliosis and astrogliosis.

IHC in the cerebral cortex disclosed intracytoplasmic granular and dense inclusions immunoreactive with AT8 antibodies (tau-P Ser202/Thr205) in neurons reminiscent of pre-tangles and oligodendroglial round inclusions or coiled bodies (Figure 4A–C). Abnormal inclusions were stained with tau-5 and anti-4Rtau antibodies (Figure 4D–F) but were negative with anti-3Rtau antibodies. Both neuronal and oligodendroglial deposits were decorated with several phospho-specific anti-tau antibodies, including PHF1 (tau-P Ser396/Ser404), tau-PThr181, tau-P Ser262, and tau-P-Ser422 (Figure 4G–N). However, the tau C3 antibody, directed to truncated tau at Asp421, immunolabeled only scarce neuronal inclusions (Figure 4O). No immunostaining was obtained with anti-tau N Tyr29.

Neuronal inclusions were abundant in the parietal cortex but dramatically decreased in number in the frontal, temporal, and occipital cortices. The paleocortex and archicortex were immunonegative, except for very rare positive neurons in the left subiculum, piriform cortex, and hippocampus. After assessing several sections stained with anti-tau antibodies, the striatum, thalamus, and mesencephalon barely showed a single tau-positive neuron. Oligodendroglial inclusions occurred in the subcortical white matter of the parietal cortex and corpus callosum, as well as some isolated coiled bodies seen in the thalamus (Figure 4C,F,I,L,N). Neuropil threads were seldom observed.

Neuronal and oligodendroglial inclusions in the cerebral cortex were also decorated with phospho-specific anti-MAP2 antibodies (MAP2-P Thr1620/1623), with the same pattern and distribution as seen with anti-phospho-tau antibodies (Figure 5A–C).

Antibodies directed against CK1δ immunolabeled small and granular neuronal inclusions of tau-affected areas, with only occasional presence in large and dense protein accumulations, (Figure 5D,E). Immunohistochemistry to phospho-kinases SAPK-JNK-P (Thr183/Thr185) and p38-P (Thr180/Tyr182) disclosed granular inclusions in affected neurons and oligodendrocytes (Figure 5F). Finally, anti-ubiquitin antibodies identified abnormal deposits in neurons and glial cells, and anti-P62 antibodies in neurons in tau-affected areas (Figure 5G–I).

Tau deposits in the cerebellum deserve particular attention. The cerebellar hemispheres were almost deployed of Purkinje cells, although small clusters were found in the vermis. The remaining Purkinje cells showed fine granular or diffuse AT8- immunoreactive deposits in the globose cytoplasm and rarely in proximal dendritic branches (Figure 5J,K).

The localization and distribution of neuronal degeneration, identified by the reduction in the number of neurons and spongiosis, and the presence of tau pathology are illustrated in Figure 6. The areas of neuronal degeneration (Figure 6A) are less extensive than those with tau pathology (Figure 6B). A quantitative study revealed the mean rations of AT8/NeuN-immunoreactive neurons in the following gyri: marginalis layers II–III: 10%, layers IV–V: 7%; ectomarginalis II–III: 16%, IV–V: 12%; ectosylvius II–III: 17%, IV–V: 9%; and intersylvius II–III: 0, IV–V: 0 (no AT8-immunoreactive neurons). No similar counts were made in the cerebellum as Purkinje cells are not stained with anti-NeuN antibodies. Purkinje cells massively died out, as revealed in sections stained with hematoxylin and eosin.

No β-amyloid deposits were seen in any region. TDP-43 antibodies did not identify abnormal deposits in neurons, glial cells, and neuropil.

#### 3.3.3. Ultrastructural Study

Ultrastructural study confirmed the neuronal damage and the associated astrocytosis (Figure 7A). We observed degenerating neurons with dense aggregates of lipofuscin and abnormal fibrillary structures in the perikaryon and its processes (Figure 7B,C). Moreover, astrocytes presented a vacuolated cytoplasm with abnormal filaments. The capillaries were edematous.

#### 3.3.4. Genetics

We analyzed the genomic sequence of the *MAPT* gene of all the animals studied to identify a potential causal mutation that could explain the development of primary tauopathy. We employed targeted genomic enrichment coupled with next-generation sequencing and variant calling. Sixteen samples were used, including the study case and other neurotypical cats. The *MAPT* locus was successfully enriched with coverage above 90% in all instances except one and a mean depth of coverage ranging from 4 to 21.8 among those 15 samples (Appendix A). GATK and VarDict were used for variant detection, resulting in 1560 and 2360 identified variants, respectively (Appendix A). Among these, 1436 positions were called by both methods. Subsequently, we focused on variants exclusive to our case individual, also leveraging the cat’s genetic diversity data from ENSEMBL, identifying 3 and 37 private mutations using GATK and VarDict, respectively. All of these variants were annotated as intronic using Annovar variant annotation. In conclusion, the coding regions of *MAPT* appeared to be well preserved in our study.

## 4. Discussion

Abnormal accumulation of hyperphosphorylated tau in neurons or glia is the hallmark of tauopathies. In humans, it has been described as a group of these diseases in which the alteration of tau is the primer driver of the neurodegenerative process (the so-called primary tauopathies) [12,27]. In animals, these pathologies are not fully characterized yet. The current study describes for the first time the case of a cat that develops tau pathology without β-amyloid deposits in a pattern that selectively affects the frontal, temporal, parietal cerebral cortices, and cerebellar cortex.

Tau protein is part of a group of proteins known as microtubule-associated proteins (MAPs), which are essential for the developing kitten brain [28]. However, the formation of abnormally phosphorylated tau can lead to a gain-of-toxic function and characterize many neurodegenerative diseases [12]. Old cats may develop a syndrome known as Feline Cognitive Dysfunction (FCD) or feline dementia, in which behavioral and cognitive alterations emerge with age-related tau and Aβ deposition, and neurodegenerative changes (e.g., cerebral atrophy, neuronal loss, ventricular enlargement, vascular changes, etc.) [23,25,29,30,31]. The main signs of FCD are summarized by the acronym DISHA: Disorientation, alterations in Interactions with owners, other pets, and the environment, Sleep-wake cycle disturbances, house soiling, and changes in Activity [25,29,30,31]. Apart from the age-related tau pathology, no other feline tauopathies have been described. Here, 16 feline brains aged from 1 to 21 years and distinct neurological symptoms were retrospectively evaluated to study the development of tau pathology.

In one cat aged 18, we found neurofibrillary pre-tangles mainly in the hippocampus and β-amyloid diffuse plaques in the neocortex. The coexistence of both tau and β-amyloid deposition confirmed a spontaneously age-related secondary tauopathy similar to the ones that other authors have previously reported [20,21,22,23,24,25]. Interestingly, another cat of 10 years of age presented a slowly progressive neurodegenerative disease that, in the post-mortem examination, was revealed as a unique primary 4R-tauopathy characterized by severe degeneration of both cerebral cortex and Purkinje cells in the cerebellum. In this case, the age at onset (4 years) and the neurological manifestations (progressive loss of balance, difficulty to walk, hypermetric ataxia of all four limbs, and intention tremor) differ from those of FCD. Moreover, when the disease progressed, the cat presented disorientation, tetraparesis, hypertonicity in the four extremities, lack of menace response, and occasional episodes consistent with epileptic seizures that defined the advanced stages of the disease. Due to the poor prognosis, she was euthanized at the age of ten years.

Neuropathological lesions were characterized by severe cortical neurodegeneration, mainly of the parietal cortex and cerebellum, together with the 4R-tauopathy principally involving the neurons and oligodendrocytes of the parietal cortex. Clinical manifestations correlate with the neuropathological lesions in the cerebral cortex and cerebellum. Indeed, some reports have associated hyperphosphorylated tau with seizure activity in cats [20,21,32], as we show in the present case. The lack of β-amyloid deposition also differs from common neuropathological changes in FCD. Regarding tauopathy, Poncelet et al. [24] reported two cats aged 19 years that also suffered from 4R-tauopathy involving neurons and oligodendrocytes and lacked β-amyloid deposits. However, these animals only presented a decreased mental status, and the distribution pattern of the lesion was found in the neocortex without the involvement of the cerebellum [24].

Tau deposits in the cat with a primary tauopathy are granular and pre-tangle-like in neurons and round or coiled in oligodendrocytes. The distribution of tau pathology largely predominates in the parietal cortex, being less abundant in other areas of the cerebral cortex and scarce if present in the paleocortex and archicortex, thalamus, striatum, mesencephalon, and cerebellum. Deposits are recognized with several general and specific phospho-site antibodies (4Rtau, tau5, AT8, PFH, tau-P Thr181, tau-P-Ser 262, tau-P Ser 422) and rarely with truncated tau at Asp421 (tau-C3). Phosphorylated MAP2 is also a component of abnormal neuronal and oligodendroglial deposits. These tau deposits are associated with increased granular expression of active tau kinases p38-P Thr180/Tyr182 and SAPK/JNK-P Thr138/Thr185, thus suggesting that abnormal tau hyper-phosphorylation depends, at least in part, on the activation of specific kinases, as seen in human tauopathies [33,34]. In the case of the active form of the glycogen synthase kinase-3 (GSK3) we did not find any cell with alterations of this protein despite it being described in abnormal tau deposits in the rare 4R-tauopathy of old cats [24]. Stress-induced protein kinase CK1-δ is also expressed in cytoplasmic granules in our case. CK1-δ is a marker of granulovacuolar degeneration [35,36], a neurodegenerative change linked to autophagy accompanying tau pathology [37]. Finally, the expression of ubiquitin and p62 associated with abnormal tau deposits suggests proteasomal and autophagy activation with impaired degradation of ubiquitinated proteins [38,39].

In the telencephalon, neuronal degeneration, as revealed by the loss of neurons and spongiosis, was wider than the presence of tau pathology in the upper cortical layers, thus suggesting that tau pathology precedes neuron loss and is subsequently accompanied by reactive astrogliosis. This scenario is further supported in the cerebellar cortex, where tau pathology occurs in the few remaining Purkinje cells of the devastated Purkinje cell layer, whereas baskets are well preserved.

As one of the causes of primary tauopathies in humans is genetic, with mutations in genes encoding the proteins that promote tau accumulation [12,27,40], we analyzed the microtubule-associated protein tau (*MAPT*) gene sequence of all cats. Genomic enrichment, next-generation sequencing, and variant calling identified 3 and 37 private mutations using GATK and VarDict, respectively. All these variants were annotated as intronic using Annovar variant annotation. The coding regions of *MAPT* were well preserved in the cat, with the primary tauopathy rejecting a possible causal mutation related to this gene. However, other genes in humans have been robustly associated with tauopathies that have not been studied in this cat, so we cannot discard a possible genetic origin of the disease.

## 5. Conclusions

To our knowledge, this is the first case reporting a primary tau proteinopathy with severe cerebral and Purkinje cell degeneration in a cat showing clinical signs at a young age (4 years old). They were characterized by hypermetric ataxia of the four limbs, bilateral episodic loss of balance, and intention tremor that progressed slowly to a severe disoriented mental status, non-ambulatory tetraparesis, generalized hypertonicity, lack of menace response in both eyes, and episodes consistent with epileptic seizures. Nevertheless, we only described one case, and we will need more research focused on the pathology of tau in veterinary medicine. Clinical veterinarians and pathologists should consider tauopathies in the differential diagnosis of neurodegenerative disease and collect clinical, neuropathological, biochemical, and genetic data to clarify the differences between human and animal tauopathies.

## Figures and Tables

**Figure 1 animals-13-02985-f001:**
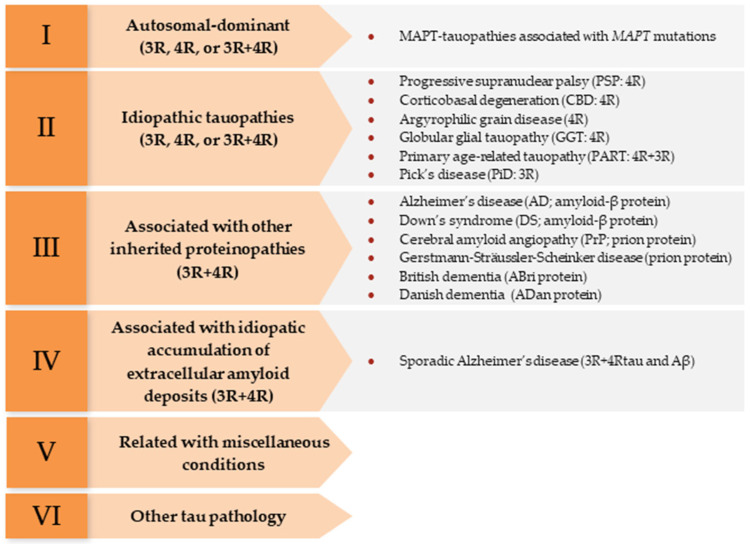
Classification of human tauopathies into six different groups depending on the anatomical distribution, the histological and cytological characterization, and the clinicopathological classification. Modified from Kovacs et al. 2022 [12].

**Figure 2 animals-13-02985-f002:**
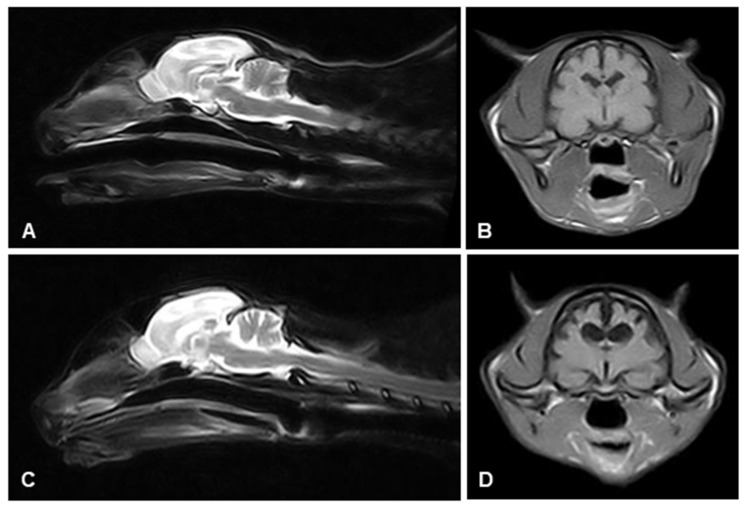
Magnetic resonance imaging (MRI) of the feline brain. Progressive cortical atrophy revealed by the prominence of all cerebral and cerebellar sulci and the dilatation of ventricles in (**A**) sagittal T2-weighted imaging and (**B**) transversal FLAIR (fluid-attenuated inversion recovery) taken at the first examination. The cerebral and cerebellar lesions were aggravated as shown in the (**C**,**D**) same sequences taken three years later.

**Figure 3 animals-13-02985-f003:**
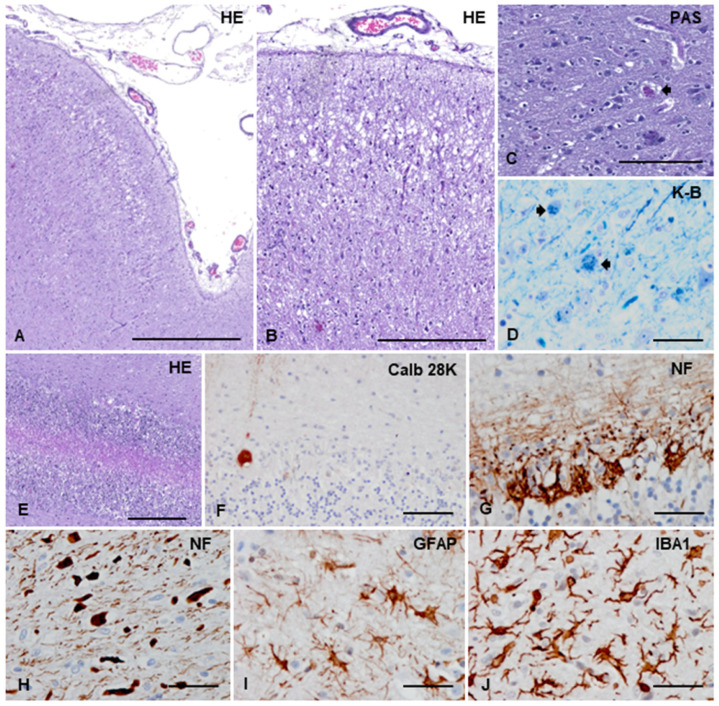
General microscopical alterations. (**A**) Marked neuronal loss mainly in cortical layers II and III with severe spongiosis in layer II and upper layer III. (**B**) Higher magnification of the parietal cortex showing neuron loss and spongiosis in the upper cortical layers and neuron loss and marked gliosis in the inner layers. (**C**) Remaining neurons containing granular PAS-positive deposits (short arrow). (**D**) Klüver-Barrera staining reveals the phospholipid component of the intracytoplasmic granular deposits (short arrows). (**E**) The cerebellar cortex shows massive loss of Purkinje cells and better preservation of granule cells. (**F**) Calbindin 28K, used as a marker of Purkinje cells, reveals the dramatic, almost absence of Purkinje cells. (**G**) Empty baskets revealed with the neurofilament (NF) immunohistochemistry appear preserved. (**H**) Axonal loss and axonal swelling are seen in the subcortical parietal white matter. (**I**) Marked astrocytic gliosis, as revealed with glial fibrillary acidic protein (GFAP) immunohistochemistry, is found in the affected cerebral cortex. (**J**) Cortical degeneration is accompanied by reactive microgliosis, as seen with the microglial marker Iba1. Paraffin sections stained with hematoxylin and eosin (HE), periodic acid Schiff (PAS), luxol fast blue-Nissl (Klüver-Barrera: K-B), and immunohistochemistry. Immunohistochemical sections are slightly counterstained with hematoxylin; bar, (**A**) = 500 µm; (**B**,**E**) = 250 µm; (**C**) = 100 µm; (**F**–**J**) = 40 µm.

**Figure 4 animals-13-02985-f004:**
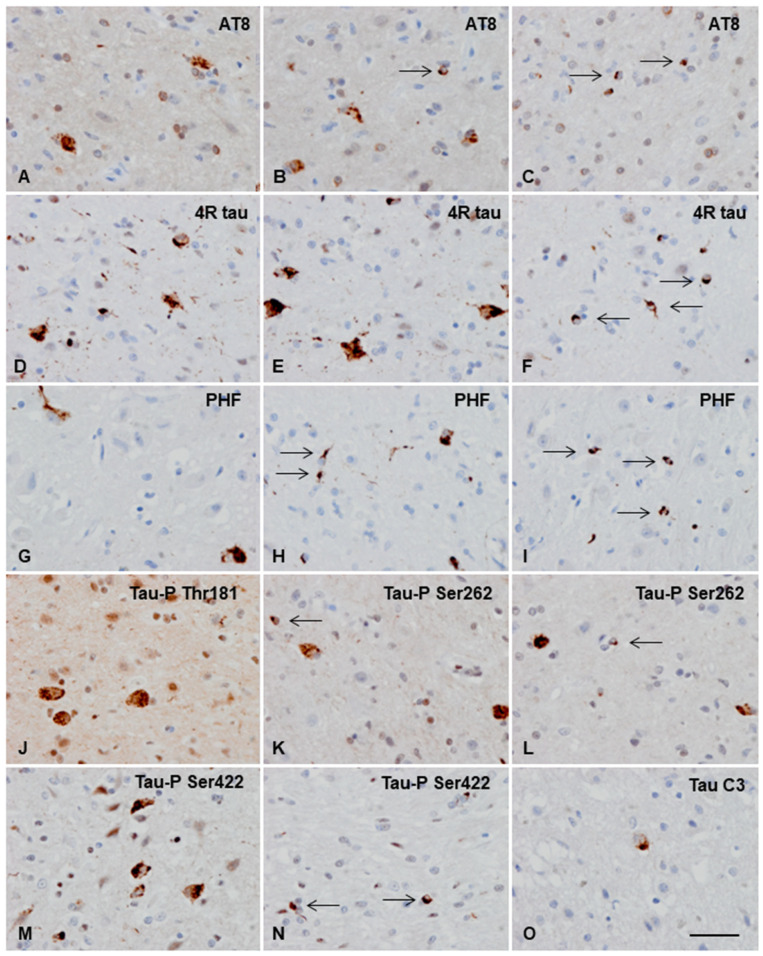
Immunohistochemical characterization of tau-immunoreactive deposits, as revealed with 4Rtau, phospho-specific (AT8, PHF, Tau-P Thr181, Tau-P Ser262, and Tau-P Ser422), and truncated tau at aspartic acid 421 (Tau C3) antibodies in the parietal cortex and subcortical white matter. (**A**,**B**,**D**–**H**,**J**,**K**,**M**) Neuronal deposits are 4Rtau and have a granular, often condensed morphology in the cytoplasm, as seen with 4Rtau and phospho-specific anti-tau antibodies (**O**). Truncated tau, as visualized with tau-C3, is found in a minority of neurons. (**C**,**F**,**I**,**L**,**N**) Glial deposits are only seen in oligodendrocytes and have a globular or coiled morphology (thin arrows). Paraffin sections slightly counterstained with hematoxylin; bar = 40 µm.

**Figure 5 animals-13-02985-f005:**
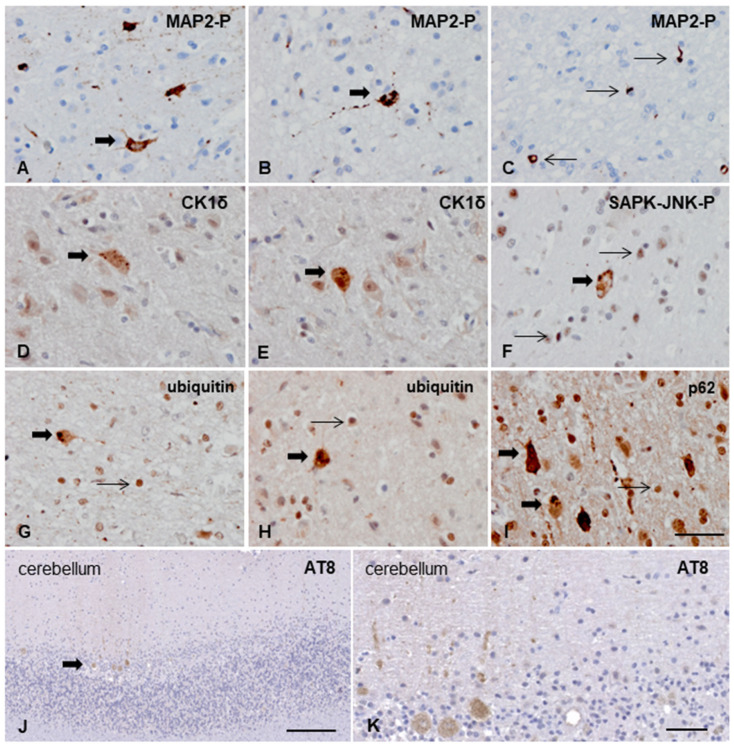
Additional deposits in neurons and glial cells. (**A**–**C**) Abnormal neuronal (thick arrow) and oligodendroglial (coiled bodies, thin arrows) inclusions are identified with anti-phosphorylated MAP2 Thr1620/1623 antibodies. (**D**,**E**) Granular intracytoplasmic deposits in neurons (thick arrows) are immunoreactive with CK1δ. (**F**) Neural (thick arrow) and oligodendroglial (thin arrows) inclusions are also visualized with anti-SAPK-JNK-P. Neuronal (thick arrows) and oligodendroglial deposits (thin arrow) are (**G**,**H**) ubiquitin and (**I**) p62 immunoreactive. (**J**) Particular AT8-immunoreactive deposits in remaining Purkinje cells in the hemispheres (thick arrow). (**K**) At higher magnification, granular or diffuse AT8 deposits are seen in the cytoplasm and scarce dendritic branches. Paraffin sections slightly counterstained with hematoxylin; (**A**–**I**), bar = 40 µm; (**J**), bar = 200 µm; (**K**), bar = 50 µm.

**Figure 6 animals-13-02985-f006:**
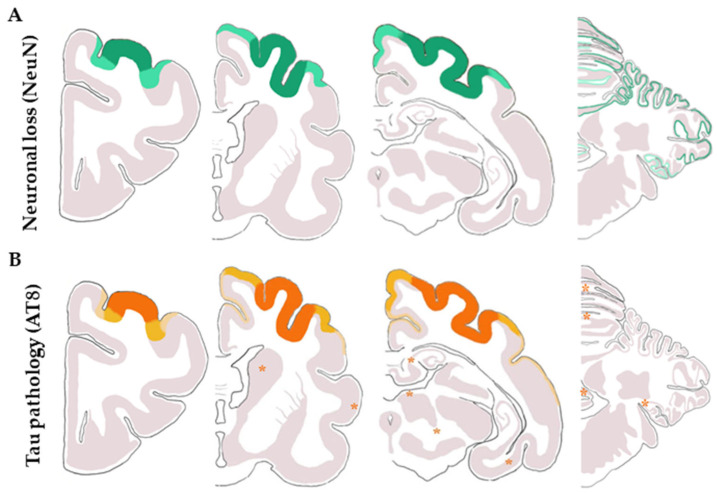
Schematic representation of coronal sections of the telencephalon and cerebellum showing the distribution of (**A**) neuronal loss revealed by NeuN considering the loss of 50−70% of neurons (dark green), a lower loss of 30–50% of neurons (light green), and apparently well preservation of neurons (grey). (**B**) AT8-positive neurons display tau pathology according to the following scale: more than 12% of neurons with tau inclusions (dark orange), between 10 and 5% of AT8-positive neurons (dark yellow), less than 5% of positive neurons (light yellow) and absence of tau pathology (grey). The presence of orange stars in the different coronal sections represents an isolated neuron with tau pathology.

**Figure 7 animals-13-02985-f007:**
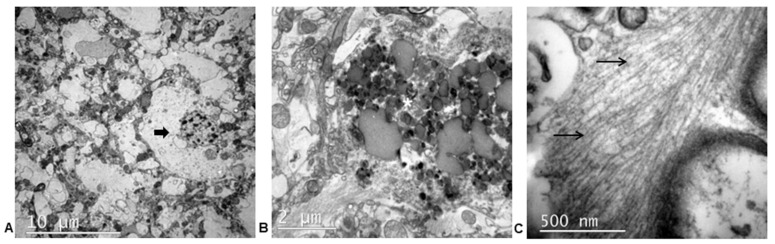
Electron microscopy of the cerebral cortex of the cat with a primary tauopathy. (**A**) Vacuolization of the neuropil and isolated astrocyte with swollen cytoplasm (thick arrow). (**B**) Intracellular lipofuscin deposits (asterisk). (**C**) Straight fibrils in the cytoplasm of a neural cell (thin arrows).

**Table 1 animals-13-02985-t001:** Antibodies used for immunohistochemistry classified as general markers and against protein deposits.

Subjects	Antibody	Supplier	Reference	Host	Dilution
General markers					
All subjects	NeuN	Millipore	MAB377	Ms	1/100
	GFAP (glial fibrillary acidic protein)	Diagnostic BioSystems	RP014	Rb	1/400
	Olig2	Merck, Darmstadt	AB9610	Rb	1/500
	Iba1	Wako	019-19741	Rb	1/1000
	NFL-P Ser473	Millipore	MABN2431	Ms	1/100
	Calbindin D28K	Sigma	C8666	Ms	1/800
	MBP (myelin basic protein)	Merck, Darmstadt	AB980	Rb	1/400
Protein deposits					
All subjects	3Rtau	Upstate	05-803	Ms	1/800
	4Rtau	Millipore	05-804	Ms	1/50
	tau-P Thr181	signalway	11107	Rb	1/50
	tau AT8, tau-P Ser202/Thr205	Innogenetics	90206	Ms	1/50
	β-amyloid	Dako	M0872	Ms	1/50
Primary tauopathy cat	tau 5	Thermo Scientific	MA5-12808	Ms	-
	tau-P Ser262	Biosource	44-7506	Rb	1/100
	PHF1, tau-P Ser396/Ser404	Dr. Peter Davies		Ms	1/250
	tau-P Ser422	Thermo Scientific	44764	Rb	1/50
	tau-C3	Millipore	36-017	Ms	1/100
	tau-N Tyr29	Millipore	MAB2244	Ms	1/200
	MAP2-P Thr1620/1623	Cell Signaling	4544	Rb	1/100
	Ubiquitin	Dako	Z0458	Ms	1/250
	p62	Transduction	610833	Ms	1/100
	CK1-δ	Abcam	ab85320	Ms	1/500
	P38-P Thr180/Tyr182	Cell Signaling	9211	Rb	1/100
	SAPK/JNK-P Thr183/Thr185	Cell Signaling	9251	Rb	1/25
	TDP-43-P	Millipore	MABN14	Rat	1/100

Ms Mouse; Rb Rabbit.

**Table 2 animals-13-02985-t002:** Breed, age, and presence of phosphorylated tau of each cat studied.

Cat Number	Breed	Age (y)	Phosphorylated Tau
1	European	1	−
2	European	3	−
3	Unknown	4	−
4	Siamese	5	−
5	European	6	−
6	European	9	−
7	European	9	−
8	European	10	−
**9**	**European**	**10**	**+**
10	Persian	11	−
11	Persian	12	−
12	Siamese	15	−
13	Unknown	17	−
14	European	17	−
15	Siamese	18	+ *
16	European	21	−

− Absence of tau pathology; + Presence of tau pathology; * Presence of β-amyloid pathology. The cat number 9, in bold, corresponds to the one with primary feline tauopathy.

## Data Availability

The data presented in this study is available on request from the corresponding author.

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
