# Peer review of "Primary Feline Tauopathy: Clinical, Morphological, Immunohistochemical, and Genetic Studies"

_animals, 2023, doi:10.3390/ani13182985_

Round 1

Reviewer 1 Report

The authors have made a wide histopathological study in a cat suffering from a tautopathy. Their results are of interest to the scientific community, especially veterinarians. However, there are several issues that have to be tackled before publication. The authors should particularly explain why they included other 15 cats in the study with different ages and clinical background. Some parts of the material and methods and results should be better presented.

SIMPLE SUMMARY

-Line 18 – I would delete “other cells of the brain” and just say “glial cells”.

- Line 19 –You mention pure and primary tautopathy several times in the manuscript but I am not sure if they can be used interchangeably. In the abstract you mention that you describe a pure tautopathy but the manuscript is entitled primary tautopathy.

- Line 20 – I would say “we describe” rather than “we described”.

- Line 21 – Including the cat you deep studied in this paper, the comparative study was done with a total of 16 or 15 cats?

I do not really think you did a comparative study among 16 cats, as they were of different ages and the clinical presentation was not always related to neurology. I would be clearer with this here in the abstract and in the material & methods and results.

-Line 21 – Please replace “clinics” by a better word such as “clinical aspects” or similar.

- Line 24 – I would rephrase this part as “Evaluation of brain tissue showed significant loss of neurons in the parietal cortex and Purkinje cells in the cerebellum”.

- Line 25 – Replace “special tests” by “immunohistochemistry”.

- Line 26 – Delete “another brain cell known as”.

- Line 28 – Add “signs” after “clinical”.

- Line 29 – You mention here and in several parts of the manuscript that the cat you studied was young. I understand that the clinical sings started when the animal was 4 but it was euthanized and the brain evaluated at the age of 10, which strictly I would not consider a young animal.

ABSTRACT

- Line 36 – Replace “abundant” by “severe”.

- Line 37 – I would be more specific with the pathological findings regarding Purkinje cells.  I would specify that it was loss of Purkinje cells in the cerebellum.

- Line 41 – Add “The” before “Genetic study”.

- Line 42 – Reading the abstract it is not clear why you included other 15 cats in your study.

INTRODUCTION

-Line 50 – Add “s” after “maintain”.

- Line 51 – “Related to” instead of “with”.

- Line 52 – This phrase reads weird, please rewrite it.

- Line 57 -Please amend the error that appears constantly in the text regarding references.

- Figure 1 – Please mention it in the text.

- Line 67 – Delete the bracket before “Based”.

- Line 87 and 92 – You say that a tautopathy was reported in 2 cats aged 19 months but then in line 92 you say that no tautopathies have been reported in aged cats. Please amend this incongruence.

-Line 95 – Add “ed” after progress

- Why did you decide to include 15 cats of different ages for comparison? Did they show neurological symptoms? Would not it be more correct to include more animals of the same age with and without neurological symptoms? What is the aim of comparing the female cat of your study with the others? Why did you do genetic studies in the other animals? This should be clarified.

- Line 100 - Replace “neither” by “none” and “analyzed” by “evaluated” or “studied”.

- Line 100 - Delete “presented”.

- Line 103 – Delete one of the full stops.

I do not think it is appropriate to give results in the introduction. Please check with the journal indications.

Table 1 is not referred in the text and should be included in the material and methods and not the introduction.

MATERIAL AND METHODS

-Here I would include a section indicating the animals used in the study.

-Table 2 is not mentioned in the text.

I do not understand if the + and – indications regarding staining in neurons and glia are results of your study or if you are showing where the staining of these markers should be seen and the morphology. If these are your results this information should not be here.  If these are indications of location of the stain you should rewrite line 130 as it is confusing.

Regarding the host in which the antibodies have been made, I would specify the meaning of Ms and Rb. I would also write “dilution” instead of “IHQ dil”

-Line 132-133 – Which other two cats were included in the ultrastructural study?

- Paragraph 162 – 166 is difficult to read as it is too long.

- Lines 168- 170 are results.

- What is the aim of using human and dog brain tissues? You do not mention your findings for comparison in the rests of the manuscript, not even in the discussion.

RESULTS

-  In 3.1. You are mixing material and methods and results. Please amend this.

-  Line 184 – Delete here the reference of the figure and add it after “ventricle was enlarged” as Figure 2 A, B.

-  I would delete Figure C, D from line 191-192 and add it as Figure 2 C, D at the end of line 195.

-  In line 199 you mention the macroscopic findings but the title states “microscopic examination and immunohistochemistry”. Pleas amend this.

-  Line 206 – The word “decorated” reads weird in this context. Please replace this word by “stained”, “labelled” or a more commonly used word depending on the context in all the manuscript.

-  Line 206 – Is the lipofuscin really making “inclusions”?

-  Figure 3 – (G) I would delete “in contrast” and “as”. (H) Replace “ballooning” by “swelling” and “is” by “are”.

-  Line 231 – Add “with” between “stained” and “tau”.

-  Line 246 – Delete “they”.

-  Line 262, 265, 266 – Delete the hyphen in oligodendroglial

-  Line 263 – Something is missing after oligodendroglial (deposits? inclusions?)

-  Line 265 and 266 – Which one is (F)? If you are showing a neuron and glial cells, please indicate which cell is the neuron with a star or an arrowhead or a thicker arrow.

-  Line 271 – Add delta after CK1 and replace “visualized” by “stained”/ “located” or a more proper term.

-  Paragraph 283-292 – Would not it be better to include this part at the beginning of the results of the section 3.2? Please make the references of the figures 6A and 6B in the text.

-  Line 300 – “represents” instead of “represent”.

-  Section 3.3 – It would be interesting to see a photographs of the ultrastructural study.

-  Section 3.5 could be presented before in the text.

DISCUSSION

-Line 341 – Delete one of the stops before cerebral.

- Lines 355-356 – I would delete “in the young cat”.

- Line 357 – I would replace “above” by “here” or the “present case” / “our case”.

- Line 357 – Delete “In our case”.

CONCLUSIONS

- Line 399 – Was it a pure or a combined tautopathy? In the abstract you say it was pure and here that it was combined. Please amend this incongruence.

- Line 400 – Delete young cat (see above) or state “in a cat showing clinical signs at a young age (4 years old)” or similar.

- Line 402 – Add “ed” after progress.

There are minor changes to make. There are some words that are not commonly used in this conext (such as decorated) and should be replaced by better terms. 

Author Response

Dear reviewer, 

Thank you very much for taking the time to review this manuscript. Please see the attachment to find a detailed response to your comments. 

Best regards

Reviewer 2 Report

In the current study, a unique and intriguing case of a neurodegenerative disease in a cat has been documented, shedding light on the rare occurrence of pure tauopathies in non-human species. In this instance, a six-year history of progressively worsening mental status and gait abnormalities was observed in the cat. Diagnostic imaging revealed widespread cortical atrophy, indicative of severe brain degeneration.

Due to the poor prognosis, the cat was euthanized at the age of ten. Detailed post-mortem examinations of brain tissue unveiled substantial neuronal loss, accompanied by noticeable spongiosis (formation of vacuoles) and a moderate increase in reactive gliosis (response of glial cells to injury) within the parietal cortex. Similar pathological changes were observed to a lesser extent in other regions of the cerebral cortex and in the Purkinje cells of the cerebellum. Immunohistochemical techniques revealed the presence of a specific type of tauopathy known as a 4R tauopathy, characterized by the presence of granular pre-tangles in neurons and coiled bodies in oligodendrocytes (a type of glial cell).

Various antibodies targeting specific phosphorylation sites on tau protein were used to detect tau deposits. These deposits were closely associated with heightened activity of tau kinases, specifically p38-P Thr180/Tyr182 and SAPK/JNK-P Thr138/Thr185. Interestingly, genetic analysis revealed that the coding regions of the MAPT gene, which encodes tau protein, were well-preserved and showed no significant alterations.

Notably, this particular case represents the first reported instance of a pure 4R tauopathy in a young cat, characterized by severe degeneration of both cerebral cortex and Purkinje cells in the cerebellum, and presenting with distinct neurological symptoms. I have some small issues.

1.     Why authors have not compared the various immunohistochemistry with the controls, so that readers can understand the severity of disease.

2.     In line 198, there is a typo. I think its IHC and not IHQ.

Author Response

Dear reviewer, 

Thank you for your valuable time and effort in reviewing our manuscript. Please find the detailed responses below to each of your comments, along with the revised manuscript in the re-submitted files.

  1. In our case, the controls we used were to validate the experimental protocol and be sure that the antibodies were correctly working and immunolabeling the desired protein. We have specified it in the new subsection "2.1. Animals and control samples" of material and methods (line 108). However, we agree with you and we could make the severity of the disease more understandable. Accordingly, throughout the manuscript, we have better explained the pathology of the other cats and compared it with the primary tauopathy in the new subsection 3.1. of results (line 198) and in the discussion.
  2. Thank you very much for pointing this out. We have replaced IHQ with IHC (line 137). 

Best regards

Reviewer 3 Report

Dear authors,

Thank you for submitting this interesting study. The paper is well-written and structured. However, in my opinion, there are some shortcomings in regard mainly to the materials and methods section. Below I have provided the differents issues and made a few suggestions.

1) Throughout the manuscript, there are several issues with the figures' and tables' citations (L57, 101,116, 184, 191, 231, 232, 235, 237, 255, 259, 272, 276, 285, 325).

2) I would suggest adding a sampling section to make it more straightforward how you have managed the different samples for the different techniques (considering the first examination and those made three years later)

3) The materials and methods have not mentioned all the techniques mentioned under the clinical history within the result section.

4) I would suggest adding a few microphotographs to the ultrastructural study

 Best Regards

Author Response

Dear reviewer,

Thank you for your valuable time and effort in reviewing our manuscript. Please find the detailed responses below to each of your comments, along with the revised manuscript in the re-submitted files.

  1. Thank you for pointing this out. We have amended the issues with the figures' and tables' citations throughout the manuscript. 
  2. We think that this is an excellent suggestion. We have moved the prior subsection “2.4. Control samples” and changed it to “2.1. Animals and control samples” (line 108). We have added all the information about the animals and positive controls used in the study. 
  3. Thank you for mentioning it. We have added a subsection in material and methods to amend it (2.2. Magnetic resonance imaging, line 131).
  4. As suggested by the reviewer, we have included a figure with the main findings of the ultrastructural study (Figure 7, line 365, referred to in the text in line 361).

Kind regards

Round 2

Reviewer 1 Report

The manuscript has been considerably improved. I mainly have some suggestions to better present the results. I would present the results following the same order as you have explained your material and methods. I would start with the clinical signs, then with the macroscopic findings and then with the results of the histopath and immunohistochemistry. This is the line normally followed in pathological studies. It is weird to find first the paragraph about tau pathology and then the clinical history and the macro/micro findings. If you have made a comparative study, also mention the clinical signs of the other cats and then focus on your deeply studied cat. After this I would explain the macro findings and then the histopath and the immunohistochemistry, including here what you mention in 3.1.

Line 111- Delete “for” after “caring”.

Line 221 – Replace “was visited” by “presented”.

Line 360-367- Thank you for including these EM photos, these are beautiful. Nevertheless, I think that it would be nice to indicate (with arrows, stars etc) in the photographs what you mention in the figure legend (e.g. where the astrocyte with the swollen cytoplasm is or which is the lipofuscin) as most readers probably will have difficulties in understanding EM images.    

Author Response

Dear reviewer,

We appreciate your valuable time and effort dedicated to reviewing our manuscript again. 

Thank you for the suggestion to better present the results. We agree that it makes more sense to mention first the clinical signs of all cats and then explain tau pathology. Therefore, we have added a section called "3.1. Clinical alterations related to tau pathology" in which we mention the clinical alterations that could be consistent with a neurodegenerative disease. Then we explain the tau pathology in section 3.2. and focus on the primary tauopathy in 3.3.

We have deleted the word you suggested in line 110 and replaced “was visited” with “presented” in line 229. Additionally, we have indicated with signs (arrows and asterisk) the alterations of the EM images (Figure 7) to make the study easier to understand for all readers. 

Best regards